# “How Do I Exist in This Body…That’s Outside of the Norm?” Trans and Nonbinary Experiences of Conformity, Coping, and Connection in Atypical Anorexia

**DOI:** 10.3390/ijerph20021156

**Published:** 2023-01-09

**Authors:** Erin N. Harrop, Hillary K. Hecht, Vern Harner, Jarrod Call, Brendon T. Holloway

**Affiliations:** 1Graduate School of Social Work, University of Denver, Denver, CO 80208, USA; 2Gillings School of Global Public Health, University of North Carolina at Chapel Hill, Chapel Hill, NC 27599, USA; 3Harborview Injury Prevention & Research Center, University of Washington, Seattle, WA 98104, USA; 4School of Social Work and Criminal Justice, University of Washington Tacoma, Tacoma, WA 98402, USA

**Keywords:** eating disorders, gender, transgender, nonbinary, body image, atypical anorexia

## Abstract

Addressing eating disorders (EDs) within trans and nonbinary (TNB) populations is a growing concern, as TNB individuals are two to four times more likely to experience EDs than cisgender women. This study explored the lived experiences of TNB people with atypical anorexia by examining how gender identity impacted experiences of ED illness and (potential) recovery. Nine TNB adults with atypical anorexia were followed for one year and completed semi-structured, in-depth, longitudinal qualitative interviews at baseline, 6 months, and 12 months. Interviews were coded using Braun and Clark’s thematic analysis procedures. Four themes, along with subthemes, emerged regarding the intersection of gender identity and ED experiences: (1) Conforming, (2) Coping, (3) Connecting, and (4) Critiquing. In Conforming, participants highlighted how societal pressures around gender contributed to ED vulnerability. In Coping, participants explained that their EDs represented attempts to cope with the overlapping influences of body dissatisfaction, gender dysphoria, and body disconnection. In Connecting, participants described ED recovery as a process of connecting to self, others, and communities that welcomed and affirmed their diverse identities. In Critiquing, participants described how current ED treatment settings were often unwelcoming of or unprepared for non-cisgender patients. Overall, participants viewed their EDs as intricately linked to their gender identity and experiences of social pressure and discrimination. This study suggests the need for targeted ED prevention and intervention efforts within TNB communities, and the ethical imperative to meaningfully address the needs of TNB patients in ED treatment settings.

## 1. Introduction

Eating disorders (EDs) are complex, multidimensional illnesses that are impairing and deadly, with patients suffering from EDs showing significantly increased mortality, decreased quality of life, increased healthcare costs, and a growing number of years lived with disability [1]. Although atypical EDs are not considered full syndromes, they result in similar symptoms, comorbidities, and levels of impairment [2,3]. For example, according to the *Diagnostic and Statistical Manual of Mental Disorders, Edition 5 (DSM-5)* [4], individuals with atypical anorexia nervosa (AAN) have all of the symptoms of anorexia nervosa except a body mass index (BMI) of less than 18.5. However, these individuals present as medically and psychologically compromised as those with lower-weight anorexia [5,6]. Unfortunately, in the United States (U.S.), though AAN is more prevalent than anorexia nervosa, fewer people with AAN are receiving treatment [7]. It is also important to note that the DSM-5 definition of AAN differs from that of the International Classification of Diseases (ICD-10) definition [8], which is broader and includes more diverse presentations than does the DSM-5. This paper deals explicitly with the DSM-5 category of AAN.

Patients with EDs face many barriers when trying to access care. In a recent systematic review, Ali and colleagues [9] found that the most frequent barrier to ED treatment was stigma and shame, often due to stigmatizing attitudes toward individuals with EDs [10]. Stigma research demonstrates that ED sufferers are often blamed by others (e.g., providers, family, and the general public) for their disorder with others perceiving them as attention-seeking [10,11] or merely viewing the ED as a lifestyle choice rather than an illness [12]. Other barriers to treatment include patient denial, minimization, and/or lack of awareness of the severity of their ED [9]. To wit, some individuals with EDs do not consider their disordered eating to be problematic [13,14], preventing them from seeking treatment. Finally, for those living in countries like the U.S. without universal healthcare coverage, ED treatment is often notoriously expensive, creating financial barriers, and disproportionately impacting those who are low-income, uninsured, or underinsured [9].

### 1.1. EDs and Treatment among TNB People

Population size estimates of transgender and/or nonbinary (TNB) individuals in the U.S. vary from 0.39% to 5.0% of the national population and have generally increased over time [15,16]. As many official records and national surveys in the U.S. do not collect information about gender modality, and TNB people in the U.S. continue to be stigmatized, underestimations are likely [16]. Concerns about EDs and treatment challenges are particularly salient for TNB individuals, as they experience EDs at higher rates than cisgender people [17,18]. This trend has been observed across different age groups, with research indicating that over 4% of transmasculine and transfeminine adolescents between the ages of 10 and 17 have been diagnosed with an ED, over three times the rate of cisgender girls, and over six times that of cisgender boys [19]. Similarly, 15.82% of trans college students reported a past-year ED compared to 1.85% of cisgender heterosexual women and 0.55% of cisgender heterosexual men [17].

Many TNB people experience body image concerns that intersect with their gender and size identities in ways that are distinct from cisgender experiences. A scoping review of TNB adolescent ED literature showed that three-quarters of reviewed case studies (*n* = 16) documented that food restriction and compensatory behaviors were used in attempts to delay pubertal development or achieve physical characteristics more aligned with their gender identity [20]. Furthermore, ED symptoms have been noted to develop among TNB populations pursuing androgynous presentations [21] and among some transgender women seeking to achieve feminine body ideals [22,23]. However, despite the elevated prevalence of and unique treatment considerations for EDs among TNB people, little research examines best treatment practices among this population [24]. Indeed, a community-based review of the largest ED treatment programs in the U.S. found that fewer than two-thirds included regular trainings on TNB-affirming treatment [25].

Current ED assessment tools may also inadequately consider TNB-specific needs [26]. This lack of research about TNB people’s experiences with EDs and treatment programs can lead to negative consequences for TNB clients who report that ED providers often overlook the relationship between gender dysphoria and body image, leading clients to believe that their treatment was ineffective and, at times, harmful [27]. As such, additional research is needed to examine how TNB identities impact ED experiences and treatment, and the role of TNB-affirming care.

### 1.2. Weight Stigma

Weight stigma is a form of intrapersonal, interpersonal, and systemic discrimination that values thinner bodies over larger bodies, contributing to negative social, financial, educational, and health outcomes [28,29]. Weight stigma is marked by negative stereotypes about the health, interests, and behaviors of people in the fattest bodies [30,31,32]. Weight stigma is also intertwined with intersecting forms of discrimination [33] including sexism, racism, ableism, and cisgenderism [34,35]. It is also worth noting that weight stigma work is often steeped in controversy as researchers and clinicians walk the thin line between responsible public health practice (that acknowledges the associations between higher weight and some adverse health outcomes) and stigmatization. Relatedly, internalized weight stigma involves negative self-beliefs about one’s own weight, often leading to weight dissatisfaction or weight loss attempts [36]. While distinct from body image, internalized weight stigma incorporates social norms about the value of body sizes [36], which can be a hallmark of ED experiences.

In ED contexts, weight stigma can negatively impact patients’ treatment access [37,38], diagnosis accuracy [39], quality of care [40], and frequency of disordered eating behaviors [41,42] by overly focusing on patients’ weight, while ignoring important behavioral, psychological, and medical concerns. Weight stigma in healthcare providers can further dissuade and harm patients, particularly patients with EDs and strong internalized weight bias [43,44]. Researchers have critiqued the ED field as perpetuating weight stigma through therapy models, provider interactions, and visual marketing materials that appear to focus care towards the most privileged patients [45].

Because of strong social expectations about gender norms in body presentation [46], weight stigma impacts people’s ability to present and perform their gender and to be perceived as feminine, androgynous, masculine, or in some other way. This is particularly salient for TNB people who do not embody cisgender privilege. For TNB people, weight stigma is an important consideration not only for gender presentation, but also for how they interact with healthcare systems. For example, in one study, TNB participants viewed provider weight stigma as a barrier to care, and as an intersectional stigma that put their health at risk [47]. In another study, TNB individuals with uteruses reported avoiding contraceptive medication due to fear of weight gain [48].

TNB patients with EDs also face weight-related barriers when trying to access gender-affirming care. In addition to harmful barriers to gender-affirming care (e.g., psychological screenings, requiring a gender dysphoria diagnosis for insurance to cover care), insurance companies and surgery providers often use BMI as a screening criterion, thus denying people in larger bodies access to gender-affirming surgeries or other care. Extant scholarship has shown that BMI requirements not only limit TNB individuals’ access to gender-affirming surgeries, but also negatively impact TNB people’s health and safety [49]. These systems enact weight stigma by mandating weight loss or a prescribed body weight threshold as a prerequisite to receiving life-saving care [49]. While gender discrimination is already tied closely to race, ability, and class, these policies add weight discrimination towards enforcing transphobic policies [34,50]. In response, weight stigma researchers are beginning to include and center the experiences of TNB people in research [51]. These complex intertwined layers of weight-based and gender-based discrimination can have the effect of othering multiply marginalized members of our communities, further reinforcing untrusting relationships with formalized healthcare.

While it is clear that EDs (including atypical EDs such as AAN) and weight stigma, pose significant health problems for the TNB community, little research has addressed the experience of TNB individuals with EDs, particularly when those individuals are of a higher weight. To address these concerns, our study used qualitative data analysis to examine the experiences of TNB persons with EDs and asked the following research questions:How did participants’ TNB gender identities impact their experiences of their EDs?How did participants’ TNB gender identities impact their experiences of treatment and/or attempts to recover?

## 2. Materials and Methods

### 2.1. Study Design, Recruitment, Screening, and Participant Characteristics

This study involved a subgroup analysis of the first author’s dissertation study [40]. In the parent study, 39 English-speaking adults with histories of AAN were enrolled in a mixed methods study over the course of one year. Participants were recruited through flyers at ED treatment centers and social media groups, provider referral, snowball recruitment, and potential participants directly contacting the researcher. Potential participants completed a brief screening appointment with the first author, a mental health evaluator and medical social worker, to ensure that participants were not acutely suicidal, acutely psychotic, or currently enrolled in inpatient care. The diagnosis was confirmed using the Eating Disorders Assessment for DSM-5 [52].

Of the 39 individuals enrolled, nine identified as TNB and were included in this subgroup analysis. Of note, five identified as TNB at the beginning of the study, and four additional participants identified as TNB before the study’s end a year later. Six participants identified as receiving public assistance before the age of 18. Demographic characteristics can be found in Table 1.

### 2.2. Ethics Considerations

All study procedures were approved by the University of Washington Institutional Review Board, STUDY00003158. All participants provided written informed consent. Study data were collected and managed using Research Electronic Data Capture [53], a secure web-based application designed to support data collection.

Individuals completed semi-structured interviews at baseline, 6 months, and 12 months. All nine TNB participants completed all 3 qualitative interviews, leading to 27 interview transcripts. Interviews ranged from approximately 1.5 to 4 h. All interviews were audio-recorded. All interviews were conducted by the first author in a private location chosen by the participant (i.e., the researcher’s or participant’s work office, the participant’s or researcher’s home, and online using Zoom).

The baseline interview focused on the development of participants’ EDs. The 6 month interview included a diagnostic DSM-5 interview to establish time frames for ED symptoms and diagnoses, followed by a semi-structured interview regarding healthcare experiences, treatment seeking, and ED treatment. The 12 month interview addressed participants’ experiences of recovery attempts and potential (re)lapses. Additionally, two weeks prior to each interview, participants received an “art prompt,”, during which they were invited to create a piece of art in response to the main topic of the interview. The art prompt was intended to prime participants for reflecting on their experiences, as well as to enable participants to convey aspects of their experience that may be difficult to verbalize. For example, in preparation for the first interview, participants received the following prompt: “Prior to the interview, please take some time to sketch out (sketch, draw, collage, paint, whatever you want!) a picture of your illness story with your ED. Consider any turning points or poignant moments that come to mind in the development of your illness”. Participants were invited to bring the art to the interview and discuss its meaning.

### 2.3. Data Analysis

The interviews followed a narrative inquiry approach. Thematic analysis followed procedures outlined by Braun and Clarke [54]. In Step 1 (data immersion), all interviews were transcribed, validated by the researcher and participant, and de-identified (all names are pseudonyms). The research team read through all transcripts, recorded notes, and marked segments that specifically pertained to gender identity. When it was unclear whether a segment related to gender, passages were discussed by the research team; if a consensus was not reached, passages were retained for analysis to avoid potentially discarding relevant data. In Step 2 (generating initial codes), the research team re-read through all transcripts and generated potential codes related to gender (42 codes). Initial codes were combined, condensed, and organized into a coding guide of 35 codes. The first author then applied this coding guide to all transcripts. In Step 3 (identifying themes), all quotes and codes were organized into themes; themes with fewer than three participants were discarded. In Step 4 (reviewing themes), all relevant quotes for each theme were collected and reviewed; a thematic map showing the relationship between themes was developed. In Steps 5 and 6 (defining themes, writing report), themes were refined, and the findings were drafted and reviewed by all members of the research team. All interview guides, art prompts, and the coding guide are available from the first author upon request.

Finally, in line with qualitative research standards, all members of the research team engaged in reflexivity practices (reflection, memos, discussion). Of note, all members of our research team identify as TNB community members in addition to holding the following other identities: White (5/5), larger-bodied (2/5), queer (5/5), ED history (2/5), neurodiverse (4/5), disabled/chronically ill (4/5), low income as a child (1/5), occupation (Assistant Professor 3/5; graduate student 2/5), and education (doctoral degree 3/5; Master’s degree 2/5). In addition to these identities, all authors identify as social work scholars. Within social work, our code of ethics states that our practice and research should “pursue social change, particularly with and on behalf of vulnerable and oppressed individuals and groups of people” [55]. As such, our approach to our research reflects the activist lens of our field.

## 3. Results

In narrating their ED journeys, four major themes arose: (1) Conforming: ED vulnerability as pressures to conform, (2) Coping: ED behaviors as coping with distress, (3) Connecting: ED recovery as connecting to self and others, and (4) Critiquing: ED treatment as unwelcoming to TNB patients. 

### 3.1. Conforming: ED Vulnerability as Pressures to Conform

When describing the initiation or resumption of ED thoughts and behaviors, participants highlighted how they were impacted by various pressures to conform within society.

#### 3.1.1. Conforming: Pressures from Societal Discrimination

Participants frequently cited that pressures from societal discrimination, including sexism, transphobia, and racism, among others, impacted their feelings about their bodies. Participants cited experiences of being “misgendered all the time” (Sisu), experiencing “street harassment” (Jessie), and not being “respected or seen as knowledgeable” due to their body presentation (Sisu). Bette cited that they lived in a “stigmatized world all the time”. Carter echoed this, saying that “it matters what you look like… people are going to discriminate against you”. Additionally, multiple participants noted that historical events, such as the election of public figures known to be homophobic, transphobic, and/or sexist, also increased levels of anxiety and disordered eating. Sisu summarized the connection between societal discrimination and their ED as follows:

“I definitely dealt with weight stigma… homophobia and transphobia… [It] was definitely not okay for me to be queer or lesbian… I still struggle with body image stuff… knowing I would be treated differently every day in my life if I was smaller… knowing my gender expression would be different if I was smaller… how I would be read would be different if I was smaller, which is one of the biggest influences for my gender dysphoria”.

In discussing their experiences of discrimination, participants also highlighted how the societal monitoring of women’s and TNB bodies increased their stress and anxiety. Mary discussed how much more monitoring she experienced as a woman, compared to when she presented more masculine, pressuring her to exert much more effort on clothing and her appearance. Jessie noted that they felt “so much pressure aesthetically” from society to present femininely and to “be at a healthy weight”. When they did not conform, “people said transphobic things”.

#### 3.1.2. Conforming: Pressures from Gender Norms, Gender Performativity, and Body Ideals

Participants cited that pressures from gender norms around thinness, fatness, and muscularity were major contributors to both body dissatisfaction and gender dysphoria. Mary, who had been encouraged to “start lifting and get some muscle, dude” when she presented as male, noted that “size is gendered… the thinner you are, the more feminine… [and] I wanted to be read more as female”. Similarly, Daisy emphasized that nonbinary individuals dealt with their own gendered appearance stereotypes: “what nonbinary looks like is a super skinny, androgynous AFAB person with short hair”.

Relatedly, Daisy noted that prior to coming out as nonbinary, pressure to perform gender was a major component of their ED:

“Overperforming femininity was very involved in my ED. I was the most feminine I’ve ever been at the sickest points [of my ED]… I felt very trapped by beauty standards… That definitely played a part in restricting… In my sickness, I was definitely trying very hard to fit into these boxes of what a woman is supposed to look like… Really striving to fit those cultural narratives and boxes of how I’m supposed to look and act as a woman”.

Participants also described that gendered body ideals influenced their experiences of their bodies. In addition to wanting to be read as feminine, masculine, or androgynous, participants also wanted to achieve beauty ideals that contributed to their EDs. Mary stated she “wanted to be a thin girl”, Carter wanted to bleach his skin and lose weight so that he could be “pretty and thin”, and Daisy was “trying to achieve this ideal of long blonde hair, thin, beautiful”. Charlie idealized two ideals simultaneously: “an ideal female 15-year-old” and “[to] look like Justin Bieber”. Nonbinary participants also endorsed wanting to have a “less curvy body” (Sisu, Bette, Jessie).

Participants explained that they felt pressure to change their bodies due to gendered patterns of fat. Mary explained that she had to be careful of how she dressed because, “Male fat patterns and female fat patterns are different… if I wear the wrong thing, someone will realize [that I am trans]”. Similarly, Jessie noted that they felt as if their nonbinary gender was less “valid” in recovery because “my breasts and my hips are getting bigger”. Bette echoed similar struggles: “it’s also hard because over my lifetime I tend to dress in a more androgynous way, and now that I’ve gotten bigger, my boobs are big, and I’ve just been wearing dresses which feels really uncomfortable”. Participants mentioned struggling to find clothing that would fit their larger bodies while feeling affirmed in their genders. Throughout the interviews, participants of all genders highlighted the disconnect between the gendered patterns of fat on their current bodies, compared to their idealized bodies. This disconnect contributed to them wanting to lose weight to be read in ways that more closely aligned with their gender identities.

#### 3.1.3. Conforming: Pressures from Weight Requirements for Gender Surgery

The last major area of pressure arose from participants seeking gender affirmation surgeries. Participants reported that because many surgeons do not operate on heavier patients (e.g., BMI > 30), they felt pressure to lose weight, which contributed to ED behaviors and cognitions. Mary stated, “My surgery date was coming up for my transition, and so I focus[ed] on my weight”. Mary felt compelled to engage in ED behaviors leading up to her surgery, because “otherwise they won’t give it to you”. However, she noted that she had “prior resolved… once the surgery was over, I wasn’t going to try to control my weight anymore”. Essentially, Mary’s ED recovery was sidelined to facilitate her surgery. Jessie noted that they were similarly worried about how ED behaviors might impact their recovery from surgery, “I’m worried with top surgery that… if I’m not eating well beforehand… it could affect the healing”. Sisu echoed similar concerns, “I might have to go out of the state to get [top] surgery… because of my size… They’re all things that make me wish that I could lose weight”.

### 3.2. Coping: ED Behaviors as Coping with Distress

When describing their experiences of having EDs as TNB individuals, participants highlighted how their EDs helped them cope with various difficult experiences, including gender dysphoria, body dissatisfaction, and body disconnection.

#### 3.2.1. Coping: ED Behaviors as Coping with Gender Dysphoria

Participants spent significant time describing the distress of gender dysphoria, which contributed to their ED behaviors. Mary describes her gender dysphoria, saying, “I would pray every night… to wake up in a body like the thin girls all around me… feeling just trapped and disgusted by the body I was in”. Sisu described how they hunched their shoulders their whole life to “minimize my breasts”. They reiterated the pain of “never being read as the gender that I feel like I am”, knowing that if they were smaller, their body would be read differently. Jessie echoed these sentiments, saying, “I hate my body… I would strongly prefer to have a smaller body, especially my hips and my chest for gender reasons”. Jessie also noted that since beginning ED recovery, their dysphoria had worsened, “because l noticed my breasts and my hips are getting bigger… at a higher weight, they’re definitely more prominent”. As a result, Jessie wanted to “relapse to make them smaller” but then quickly noted, “I don’t want to do that”. Jessie clarified that they are “trying to do things to deal with my dysphoria that aren’t going to kill me”.

#### 3.2.2. Coping: ED Behaviors as Coping with Weight Dissatisfaction

In addition to experiencing distress due to gender dysphoria, all participants reported high distress from weight dissatisfaction. Layla’s weight concern began in childhood: “[I knew] what my mom thought was a beautiful body. And, I never had that body. Always been short and stumpy”. When recounting why their ED developed, Jessie explained, “I was really concerned about my weight ‘cause when I started to go through puberty… I was a little not as skinny as [my peers], and… people made comments… I was really concerned with my weight”. Bette explained, “I thought that I was huge. I would only allow people to take photos of my head. I never ate in front of people. I keep leaving my house anticipating that I’m gonna be shamed”.

Finally, it is notable that participants described their distress from gender dysphoria and weight dissatisfaction as “very intertwined” (Mary). Mary explained this by saying, “they are really hard to separate sometimes… Is this a gender issue that is causing me grief? Or is this a body size issue?”

#### 3.2.3. Coping: ED Behaviors as Coping through Body Disconnection and Avoidance

Participants reported that they coped with this body distress through ED behaviors and disconnecting from their bodies. This disconnection manifested in body avoidance and dissociation. Charlie described their ideal body as “a brain in a jar” and a “robot”, a sentiment echoed by Bette, who also described their bodily experience in their ED as “a disembodied head with the mechanicalized body”. Both Daisy and Sisu described their EDs as “cages” or “prisons” that kept them both separated from and trapped within their bodies.

Jessie explained that they coped with their body distress by “mostly not thinking about it, not having a full body mirror, not knowing how much I weigh”. Carter described his bodily disconnection as one of a slow disappearance, saying “It went from just being like a diet… to the point where all of the weight must be gone. Like everything that I am has to disappear, because I want to disappear”. Bette shared similar sentiments, saying, “I just sort of wanted to be invisible… Let’s not acknowledge that I have a body”. Hope similarly described their experience of their ED as “a very disembodied sort of existence”.

Sisu experienced similar disconnection from their body in their ED, and explained how difficult it was to deal with the aftermath of this body avoidance once they entered recovery: “Now [in recovery]… I am in my body, and I have to deal with the impact of not seeking medical care for any previous injuries and the impact of twenty years of exercise when you are not embodied… you just don’t breathe that well when you are not embodied”. Bette expressed a similar struggle in their journey from bodily disconnection: “Part of my ED has been turning myself into basically like an armored robot, where my head and body were very separate, and my recovery has been human, having a fleshy body, and integrating my mind and body together. And it’s painful, and it feels raw and vulnerable”.

### 3.3. Connection: ED Recovery as Connection to Identity, Body, and Communities of Resistance

When describing their experiences of recovering from EDs, participants emphasized an overall theme of building connections. They described connecting to their identity, connecting to their bodies, and connecting to communities and ideologies that emphasize resistance to dominant, oppressive norms.

#### 3.3.1. Connection: ED Recovery as Connection to Identity

Multiple participants noted that connecting to their gender identity more deeply was a turning point in their ED journeys: “Being trans is a big part of my story” (Mary). Daisy summarized this saying, “Gender identity and my sexuality was really tied up in my ED… I identify as nonbinary now… when I came out felt like a big piece of my ED recovery… that was a big turning point”. Daisy went on to explain this process, “Throughout recovery, there was that systemic piece of looking at gender roles and sexuality and… diet culture… I stopped really trying to fit into the boxes [of gender], and it’s been a critical piece of my recovery”. For Daisy accepting their nonbinary identity “has felt like a big part of healing my relationship with food and movement and my body” because so much of their ED had been “wrapped up with performing femininity”. Daisy emphasized that now they felt much more “relaxed and free, connected”.

Sisu described the role of gender identity in their recovery saying, “[Recovery] means not …wearing a mask all the time…being the most free I have ever been in my body… That changed after I started realizing stuff about my gender”. When asked what had changed in their ED journey after coming out as nonbinary, Sisu explained, “There is more connection… It is easier to connect with other people… I don’t feel as lost and lonely and hopeless… I have a lot more compassion for myself. I have a lot more confidence”. Sisu clarified that although exploring their nonbinary identity had been mostly positive, they also continued to experience gender dysphoria. Jessie reported similar struggles with ongoing dysphoria, but emphasized that recovery meant finding ways to “affirm my gender [without] starving”. Specifically, Jessie mentioned they felt increased gender connection when they changed their name to their chosen name and when people “consistently us[ed] the correct pronouns”.

#### 3.3.2. Connection: ED Recovery as Connection to Body

Participants who were farther in their recovery processes also talked about connecting to their bodies. For Mary, this meant “healing my relationships with food and my body”. Sisu described their process of connection as “reclaiming my body”, and “letting my body be and listening to my body… to not push it, to teach it that I will listen”. Sisu understood their relationship with their body as a “two-way street”, and explained that this realization “made me work way more collaboratively with my body”. Recovering participants spoke of connecting to and accepting the reality of their current bodies, often in the midst of ongoing pain from gender dysphoria and weight dissatisfaction. Mary described her healing this way:

“When I started recovery and started to transition, a lot of the motivation was predicated on the notion that I would eventually be able to get to that thin girl’s body… Somewhere along the line I realized… I’m targeting a very different body. I’m targeting a body that’s, like, halfway between male and female and will be on the larger side. And, do I still want that? Yes”.

For participants who were earlier on in their ED recovery journeys, this increased sense of body connection often came with challenges of its own, such as increased gender dysphoria. For some, being more connected to their bodies, meant increased awareness of dysphoric experiences. In reflecting on their early journey, Sisu explained that their gender dysphoria actually increased (for a period) after coming out as nonbinary. Bette echoed similar bodily discontent saying, “Going from a robot body to a fleshy body… [was] very disruptive to my identity and sense of self”. Jessie identified similar struggles with their current experience of ED recovery saying, “[I hope] that I get to the point where I accept that my gender is valid regardless of how big my breasts are, but… I’m terrified of having a bigger body, [and] I’m terrified of being malnourished”. In sum, body connection and acceptance were experienced differently by participants at different points in their recoveries.

Participants also spoke of hearing conflicting messages about how to relate to their trans bodies in ED recovery. Mary eloquently summarized her double bind as follows:

“Being a trans person, the prescription for treating body dysphoria is change your body, right? Take hormones. Get surgery… Make your body however you see in your head. But the prescription for [an] ED… is the exact opposite: love yourself as you are, and don’t change a thing. It is a Catch-22… I went through a lot of struggles trying to reconcile those two ideas”.

Sisu and Jessie echoed similar struggles about how their ED may be impacted by their transition. Sisu wondered, “What are all my friends going to think if I want body sculpting to get rid of my hips? Is that going to be seen as liposuction to make my body smaller?” Sisu mourned that without these surgical procedures, they worried that they would “never be read as the gender that I feel like I am”. Jessie concluded that while they often felt torn between recovering from their ED and their ideal gender expression, they were “trying to do things to deal with my dysphoria that aren’t going to kill me”. Participants voiced ongoing concerns about how to maximize gender euphoria (and minimize dysphoria), while not falling back into ED behaviors.

#### 3.3.3. Connection: ED Recovery as Connection to Communities of Resistance

Participants also emphasized the importance of connecting to communities of resistance. While the specific communities differed, participants emphasized the importance of connecting to ideologies and communities that resisted dominant, oppressive norms (e.g., gender, health, beauty ideals, politics). Just as participants experienced pressures to conform as maintaining their EDs, participants sought to resist these pressures when seeking to recover. Sisu summarized this by saying, “[Recovery] means not conforming. A lot of my ED was connected to conforming or trying to conform”. Sisu described how recovery had changed their identity: “I have just gotten more and more radical, more and more community-focused, and more and more wanting to burn all the horrible systems to the ground”. To Sisu, resistance meant putting the blame for EDs on oppressive societal systems, “The problem is outside my body. Tak[e] the focus off of individual change, and focus on systemic change”.

Carter, who also identifies as a social activist, found this systemic lens important in his recovery, sharing this perspective-changing advice from his therapist: “every time I’d be engaging in ED behaviors, she’d be like “you are letting them win. This is them controlling you, not you controlling them. And you want to rise up against them””. Similarly, Daisy explained how they resisted dominant norms, saying, “I have lots of friends who are nonbinary. We all look totally different. It’s accepted. Everyone is fat positive… the people I’m around are not enforcing these stereotypes, or upholding them, or engaging in them at all. We are actively against that”.

As part of resisting dominant norms, participants also spoke of existing outside of norms (or “boxes”). Daisy described this concept saying, “to stop attempting to fit into that ‘woman box’, of what a perfect woman looks like, has felt like a big part of healing my relationship with food and movement and my body”. For Daisy, “removing themselves from the gender binary” offered them freedom that supported their ED recovery, and their gender identity. Charlie expressed similar sentiments: “I think a lot of times people don’t know what to do with me. Or like where to put me… If you don’t put me in any specific box, you have to sort of look at [me as] an individual”. For Charlie, part of existing outside the norms meant releasing previous engrained body ideals that had supported their ED: “As I came more into my nonbinary gender identity, over time I realized that my idea about what I should look like was eroding… when you’re not represented, there is no person that you can hold up as an ideal”. Sisu echoed these sentiments, “How do I exist in this body… that’s outside of the norm, [that] breaks all the ideals (besides being white)? …All of that [body ideals] is just thrown out of the fucking window. I don’t have any expectations that I’m trying to live up to”. For Sisu and Charlie, existing outside of the norms aided them “in being authentic”, but both acknowledged this authenticity came at a cost due to societal discrimination.

### 3.4. Critique: ED Treatment as Unwelcoming to TNB Individuals

Lastly, throughout the study, participants frequently critiqued the U.S. healthcare system and ED treatment as being unwelcoming to or unprepared for TNB individuals. Within their critiques, they pointed to basic unaffirming environments (“trans 101” issues such as misgendering, lack of TNB providers, and having to educate providers), TNB issues being unaddressed in treatment, and feeling out of place in ED treatment.

#### 3.4.1. Critique: Basic Unaffirming Environments

When discussing their interactions with providers in healthcare and ED treatment, participants noted that many providers lacked basic knowledge of issues pertinent to TNB patients: “Do I need to explain what that terminology means? How much do I need to talk about trans 101?” Participants spoke of providers who “assumed I’m straight and cis” (Jessie) and who needed to be educated about issues such as pronoun usage and nonbinary identities. Bette explained, “I’m doing so much work to just explain my lens… it’s too much effort”. Multiple participants emphasized how fatiguing it was to have to constantly educate their providers about TNB issues: “there ends up being a lot of explaining or just a lot of unintentional microaggressions… Ends up being more work than it’s worth” (Sisu).

Additionally, participants emphasized how the lack of TNB-identified providers in ED treatment was also challenging, as they yearned to have providers who shared their identities. Daisy stated, “I’ve had exclusively thin, white, cis, straight men as doctors… And, thin, straight, cis white women [in ED treatment]”. Bette similarly noted that all of their ED providers were “all skinny… cis-appearing white women… and they present as upper middle class… I’m conscious of feeling judged”. Hope mentioned feeling increased anxiety when their providers were all cisgender, thin women; Jessie questioned whether their cisgender providers could be trusted to provide good care to nonbinary patients and noted that an absence of diverse providers was “part of why I haven’t … pursued any sort of specialized [ED] treatment”. Mary wondered how her experience in treatment might be different if she had TNB providers.

#### 3.4.2. Critique: TNB Issues Unaddressed in ED Treatment

While participants noted that exploring their gender identities was a critical element in their ED recovery processes, they highlighted that TNB issues were often not addressed in ED treatment spaces. Bette struggled to engage with their providers around how challenging it was to find androgynous clothing in larger sizes: “I felt that [gender presentation] was a conversation that they didn’t know how to hold space for [and they didn’t know] why it would matter so much to me”. In reflecting on their treatment experience, Bette said, “there’s no space to talk about how gender identity or identity in general fits within the space of EDs… It is a multifaceted thing… [Treatment] just sort of missed the complexity of all of those pieces”. Charlie encountered similar struggles: “With my first therapist we talked about gender a little bit… she couldn’t quite get her mind around the concept that I don’t want to be a boy, and I don’t want to be a girl either. It just made her uncomfortable to talk about, so we just didn’t”. In reflecting on the impact of not addressing gender identity, Charlie explained, “It’s an aspect of myself and the way in which I view my body… If we’re just not talking about it, that’s ignoring an entire segment of this treatment”.

#### 3.4.3. Critique: Feeling out of Place in ED Treatment

All participants mentioned feeling out of place in ED treatment. Bette explained, “[When] I got specific ED treatment I felt… like I didn’t really belong there… The hardest part of my recovery has been… that I have to create my own space and tell people that I belong there, when I have never felt welcomed”. Multiple participants mentioned that stereotypes of ED patients as “real thin white girls” (Charlie) made it hard for them to feel as if they fit into treatment spaces. Mary explained, “At the height of my restriction, no one was saying, hey, this person has a problem… I wasn’t what you would picture as someone who was anorexic”. These feelings of not fitting in were exacerbated for participants with multiple marginalized identities, such as Carter, a Black, multiracial, queer trans person. In reflecting on his ED treatment experience, he concluded that he felt, “lonely and isolated”. Sisu, who never had formal ED treatment somberly concluded, “ED institutions have to change, or recovery is never going to be possible for the majority of the people”.

## 4. Discussion

This study examined the lived experiences of TNB people with EDs (*N* = 9) by analyzing how gender impacted their experiences of ED illness, treatment, and/or recovery. The participants were followed for one year, completing interviews at baseline, 6 months, and 12 months. Overall, TNB individuals viewed their EDs as complexly and intricately linked to their gender identity, thus suggesting the need to foster intersectionality-informed identity development in ED treatment. Relatedly, it seems important to note that in the course of this one-year study, four participants came to identify as TNB. This is likely a reflection of both historical timing—more TNB visibility in the media and increased social consciousness [56]—and the identity work in which people were engaged as they were processing and healing from their EDs. The latter highlights the importance of incorporating intersectionality-informed modalities into ED treatment for all patients—not just those for whom gender issues are identified at the start of treatment. While ED treatment often focuses on physical healing and eliminating ED behaviors, exploring the roots of their ED may lead to self-discovery and healing in other areas, such as gender identity.

Regarding the theme of conforming, participants experienced multiple societal pressures related to body image, in addition to pressures for thinness. Participants frequently discussed how challenging it was to distinguish thin idealization and weight dissatisfaction from gender dysphoria, and some questioned whether or not these experiences were ever distinct. However, the participants also emphasized the importance of having therapeutic spaces to explore these concepts, in addition to having support from ED providers in body image work and transition-related care. This finding reiterates similar findings from Cusack, Levenson, and Galupo’s study [57], which found that TNB ED patients found therapy conversations exploring gender dysphoria, diversity, and gender role expectations to be useful in ED recovery.

Additionally, participants noted that they experienced societal pressures to conform to normative gendered expectations while simultaneously facing barriers that made it harder for them to access medical transition-related services. While not all TNB persons desire or pursue gender-affirming treatments and surgeries, some do, and participants in this study noted how current BMI cut-offs for gender-affirming surgeries exacerbated their ED thoughts and behaviors. This phenomenon of being denied gender-affirming surgeries due to body size is increasingly well-documented [49]. It is important to note, however, that while the positive impacts of these surgeries on the health of TNB patients are well-documented, weight/BMI requirements are not specified in the WPATH standards of care, and such cutoffs lack empirical evidence related to surgical outcomes. Additionally, Rothenberg and colleagues [49,58,59,60,61,62] found that there are no significant differences in surgical complications or revisions between TNB individuals regardless of BMI, including those with high BMIs. This, along with existing scholarship [49,61], demonstrates that BMI should not be used to assess a TNB person’s ability to access gender-affirming surgeries.

Regarding the theme of coping, participants emphasized the role their ED played in helping them cope with body and gender dysphoria. Thus, during recovery, patients will likely need new additional coping tools to replace the role of ED behaviors [20]. Coping with gender-related distress may include affirming name changes and pronoun changes (as it did for Jessie), especially when those around the individual solicit, support, and consistently use their correct name and pronouns. Additional paths towards gender euphoria [63] include wearing affirming and well-fitting clothing (e.g., Bette, Sisu), exploring accessories, tattoos, and hairstyles, as well as using affirming sex-segregated facilities [64]. This may or may not include gender-affirming medical care including hormones or surgeries (e.g., Mary, Jessie, Sisu), or social body changes such as starting or stopping hair removal and makeup. Utilizing coping strategies that reduce body dissatisfaction and increase body euphoria that are not reliant on food restriction or thin body ideals are a form of harm reduction in safe and sustainable coping mechanisms.

In describing treatment experiences for EDs and gender dysphoria, participants highlighted mixed and conflicting messages. For instance, during the process of affirming one’s gender identity, providers may suggest that physical changes could help them achieve gender euphoria, and safely present their gender identity to others. This invitation to change and adapt their bodies (temporarily or permanently) to find joy is in stark contrast to ED treatment contexts which often suggest that intentionally changing one’s body may interfere with ED recovery. This duality is unhelpful and unactionable, lacking nuance in how people live in and evolve in their bodies over time. For example, transfeminine people who seek and receive gender-affirming medical interventions experience fewer ED symptoms, as moderated by body satisfaction [65]. Therefore, theoretical models of recovery tailored towards the complex and nuanced realities of TNB ED patients are needed.

Regarding the theme of connecting, participants highlighted the importance of connecting to communities and ideologies that supported their TNB identities while trying to recover from their EDs. Using critical consciousness to resist systemic discrimination is a protective tool in multiple therapeutic contexts [66,67]; integrating critical consciousness into ED treatment could help TNB participants shift the narrative from “my body is the problem” to “the system is the problem”. As anti-trans sentiment and enacted legislation have gained renewed energy in the 2020s across the U.S., United Kingdom, and other parts of the world, TNB folks are acutely aware of both increased visibility [68] and increased backlash. Recent threats to TNB communities include threats to TNB bathroom usage, exclusionary educational policies, efforts to prevent sports participation, employment discrimination, and barriers to gender-affirming healthcare [69]. Helping patients with EDs understand and contextualize the systemic factors that support their EDs (e.g., fatphobia, sexism, transphobia, racism, thin idealization) could help TNB patients heal from their EDs, as they learn to shift the blame away from their bodies, and onto the systemic forces acting upon their bodies.

Finally, when critiquing the current landscape of ED treatment, participants echoed much of what has been previously documented in TNB and ED healthcare literature: (1) patients struggled to find TNB-knowledgeable, accessible, affordable ED providers, (2) few ED providers shared TNB identities, and (3) basic TNB-affirming care principles (e.g., correct pronoun usage, gender options on intake forms, bathroom and housing availability) were often missing. Healthcare systems would benefit from tailored trainings to safely serve their patient populations, particularly regarding respectful communication [70] and quality improvement measures to make clinic spaces increasingly safe for TNB patients [71,72]. In particular, approaches centering the importance of upstream systemic change such as Structural Competency approaches (as opposed to interventions focused at the individual and interpersonal levels), may be most successful in creating meaningful systemic change [73]. In the U.S., TNB medical education is often relegated to one-time special topics, and not applied broadly to building ongoing clinical skills [74]. Further, even when providers receive TNB education, it may still be difficult for patients to readily identify competent providers [75,76]. Relatedly, international ED providers share many social identity demographics (thin, white, cisgender women), contributing to narrowing stereotypes in the ED field [77]. Such stereotypes may contribute to TNB patients feeling “out of place” in ED treatment spaces, as reported by the participants in this study.

While this study had many strengths, including prolonged participant engagement, high retention of TNB participants (100%), the use of longitudinal qualitative data, in-depth interviews, and TNB-identified research team members, there are several important limitations to note. First, this study utilized a subgroup analysis of all data from trans and nonbinary participants; the parent study did not exclusively focus on TNB experiences or attempt to solely recruit TNB individuals. Further studies explicitly focusing on TNB ED experiences will expand our findings. Second, our sample size was small, and while this is typical for studies employing narrative inquiry approaches (where the emphasis is placed on depth of narrative data vs. higher numbers of participants), this did significantly limit the range of TNB experiences included in this analysis. The participants in this study were largely nonbinary people, with only one trans man and one trans woman participant. Future studies of TNB populations with EDs should seek to recruit more TNB participants (across a broader range of demographic characteristics) to better represent the full spectrum of TNB experiences. Finally, much research on TNB populations has focused on a deficit- or damage-centered narrative around TNB experiences [74], and our paper is no exception. While it is important for research to illuminate the hardships faced by TNB people, damage-focused narratives fail to show a holistic picture of TNB experiences, which also contain narratives of belonging, community-building, and trans joy. While some of these positive themes were introduced in the theme of “connecting”, we hope that future TNB ED research will more explicitly focus on places of strength and joy in TNB narratives.

## 5. Conclusions

Addressing EDs within TNB populations is a growing concern. To our knowledge, this is the first qualitative study to explore the role of TNB identity in patients with AAN. Overall, participants viewed their EDs as intricately linked to their gender identity and experiences of social pressure and discrimination. This study suggests the need for targeted ED prevention and intervention efforts within TNB communities, and the ethical imperative to meaningfully address the needs of TNB patients in ED treatment settings.

## Figures and Tables

**Table 1 ijerph-20-01156-t001:** Demographic Characteristics of Participants.

Alias	Age in Years	Gender	Sex Assigned at Birth	Sexual Orientation	Race	Highest Degree	On Public Assistance as a Child?
Bette	38	Nonbinary/genderqueer	Female	Pansexual	White	Master’s	Yes
Hope	30	Nonbinary/genderqueer	Female	Queer/lesbian	White and Hispanic	Master’s	Yes
Mary	26	Female	Male	Homoflexible	White	Bachelor’s	No
Layla	38	Nonbinary/genderqueer	Female	Bi queer	White	Doctorate	No
Carter	18	Male	Female	Pansexual	African-American and White	High school	Yes
Daisy	25	Nonbinary/genderqueer	Female	Queer	White	Bachelor’s	Yes
Charlie	32	Nonbinary/genderqueer	Female	Queer	White	Doctorate	Yes
Jessie	25	Nonbinary/genderqueer	Female	Bisexual/queer	White	Some college	No
Sisu	36	Nonbinary/genderqueer	Female	Queer	White	Bachelor’s	Yes

## Data Availability

The data presented in this study are available upon reasonable request from the corresponding author. Data are not available publicly due to the confidential nature of the in-depth life narrative interviews of participants, as described in the study’s informed consent process.

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
