# Peer review of "“How Do I Exist in This Body…That’s Outside of the Norm?” Trans and Nonbinary Experiences of Conformity, Coping, and Connection in Atypical Anorexia"

_ijerph, 2023, doi:10.3390/ijerph20021156_

Round 1

Reviewer 1 Report

Major comments

1) I am very concerned about table 1. It contains sensitive information that greatly hamper anonymity. I am not sure how this complies with ethical standards.

2) The limited sample size, and the secondary analysis is a major limitation of the survey, and should be more thoroughly addressed.

Other comments

1) Minor English editing is required

2) L96 -- Reference is not in journal style format

Author Response

Please find attached our attached manuscript entitled, “’How do I exist in this body…that’s outside of the norm?’ Trans and nonbinary experiences of conformity, coping, and connection in atypical anorexia,” which my colleagues and I are re-submitting for publication in the International Journal of Environmental Research and Public Health special issue on “Health and Healthcare for Transgender and Gender Diverse Communities.”

We are grateful for the reviewers’ thorough comments and have addressed each of their concerns. We have included a response table outlining each comment from the reviewers, our response, and the revised language that has been added to our manuscript. Additionally, we have also included a document with tracked changes, highlighting all of the revisions to our manuscript. We believe that after incorporating the reviewers’ feedback our manuscript is much improved. We hope that we have addressed the reviewers concerns in a thorough and satisfactory way, and we look forward to your feedback.

Reviewer 2 Report

Dear Sir or Madam,

I thank the editors for the opportunity to review this article and the authors for their work.

In this article, the authors present the results of interviews with participants who identify as TNB and simultaneously have an eating disorder. They aim to outline the respondents’ experiences in seeking care and shed light on the interplay between TNB gender identity and eating disorders.

They hereby touch upon a so far much under-researched topic and fill a gap in the current state of knowledge. I therefore think that this article will be of interest to many readers, public health practitioners and psychotherapists alike.

To further improve the article prior to publication I suggest some revisions relating to the following aspects:  

P1Abstract: I wonder if the terminology is inconsistent, or if I am just not familiar with the terms used: Are “gender expansive”, “trans and nonbinary” and ”gender diverse” synonyms? If not, how do they differ?

P1l36: I wonder if eating disorders are first and foremost “social problems”. Why do the authors privilege eating disorders’ social effects over their effects on the development and health of the affected individuals? This perspective would merit some more elaboration.

P1ll40-42: The authors’ definition of atypical anorexia nervosa differs substantially from the criteria outlined in ICD 10. Maybe it would be helpful for European readers to acknowledge this difference and explicate that the definition of AAN follows the classification of DSM here.

P2l55: In many European health care systems patients don’t pay for their treatments, so the cost of treatment does not pose a barrier to treatment as such. I would recommend that the authors qualify their statement by adding something like “in the US” or so.

P2l58ff: I think it would be interesting to have some information on the overall size of the group of TNB people within the population. Even though the prevalence of eating disorders is high among them, TNB people with eating disorder are overall a very small group. To put the study’s findings into perspective, this should be mentioned somewhere in the introduction.

P2l88ff: The authors rightly focus on the problems related to weight stigma and justifiably argue against stigma attached to large bodies. I completely agree with this line of reasoning.

At the same time, I think it would be important to mention that not all discussion of overweight reproduces stigma. After all, obesity is a major public health challenge, whose importance is often obfuscated by debates about body positivity. I would like to recommend that the authors explicitly acknowledge the thin line between responsible public health practice and stigmatization. 

P3l96: The reference to Lee et al. is not correctly formatted.

P3l109: Does “gender expansive people” here refer to a subgroup of the “TNB people” mentioned in the next line? If not, I suggest using one term consistently.

P3l116-119: By classifying the requirements for reassignment-surgery as “arbitrary” the authors adopt an activist perspective here without a comment; in the medical community there are people who still consider these requirements as necessary and meaningful. If the authors disagree with them, I suggest explaining the disagreement, outlining why the requirements seem arbitrary and citing the respective literature.   

P3l131: The abstract mentions qualitative interviews, but here the authors speak of “secondary data analysis”. How does this go together? (I now see that it is explained later on. So I suggest omitting the reference to the secondary data analysis here to avoid confusing readers.)

P4l149: In my understanding, secondary data is defined as data that was collected for a purpose other than science (e.g. clinical care) and is then used “secondary” to answer scientific questions. After reading the explanations on page 4, I therefore think that this research is mislabeled as “secondary data”; what the authors do is performing a sub-group analysis within a bigger study, but they use primary data.

P4l154: The information on the ethics review is missing. Please also add the registration number of the ethics proposal.

P4Table1: For a European reader, the column on “Ethnicity” is a bit puzzling. I think the article would benefit from some elaboration in the difference between race and ethnicity here. In the European discourse, both terms are usually used interchangeably.

Also, please explain the abbreviation JD and outline, how “low income” was defined. And: Is DoorDash an occupation, or does it name the respondent’s employer? If the latter, I suggest changing it to a term outlining the actual work, in line with the description of the other respondents.

P4l160: Please add who conducted the interviews and where.

P5l169: I applaud the authors for this idea! Could you give some more details concerning the art prompt?

P5l194ff: I think it would be interesting to learn how the research team’s reflexivity influenced the data analysis. The article transpires a somewhat activist motivation of the research team, and I think it would be helpful to reflect on this aspect of positionality as well.

In addition, I wonder why the authors decided to reflect on the outlined aspects of identity, but not on e.g. class or occupation/education/profession.

P6l218ff: The quotation marks around Sisu’s statement are missing.  

P6l241ff: Again, the quotation marks are missing. This is also the case for many of the following citations. Please add for better readability.

P7ll268-280: I would wish for more context concerning the relevance of weight for the surgeries. Is the weight restriction on the side of surgeons motivated by medical reasons, such as the increased risk in very obese people that might discourage performing elective surgeries in general? Or are there other arguments? Or is it simply discrimination?

To understand this paragraph better, it would be helpful to learn what it means in medical terms that the respondents were “heavier”. Are we talking about a BMI of 30, or a BMI of 45?

P7l313ff: I think the observation, that ED act as a means to achieve body disconnection is a very remarkable finding and interesting on many levels. Could you elaborate this more? I would love to read more about this. Would you present more citations from the interviews?

P10l437: Is there maybe a better word for “health care systems”? I think what the participants criticize are not “systems” but elements or components of the American health care system.

P10l466: As a non-native speaker, I struggle a bit with the wording “… as a missing part of treatment”. Does this imply something other than “TNB issues were often not addressed in ED treatment settings”?

P12l530: While weight requirements might not be specified in the WPATH standards and the transgender-related literature, they are generally assumed to be valid in most fields of surgery. So I think it is not correct to claim that focus on weight for elective surgery lacks empirical evidence. See for instance the systematic review by Griffin et al. (2021): Nutrients, doi: 10.3390/nu13113775

P12l565: The wording seems to be a tick too colloquial here: “TNB folks are hyper aware…”

P12l568: In this sentence it seems something is missing, it seems grammatically off.

P13ll585-589: These two sentences seem to imply that replacing the current workforce of health care providers by non-thin, non-white, non-cisgender providers would somehow solve the problem. I think this is a mislead conclusion. I would suggest that the authors look into the concept of structural competency and see if this might offer solutions for the problems they describe that go beyond selecting health care providers along the lines of (ascribed) identities.

P14l636: I don’t understand what it means that the work was completed on the “unceded land” of the mentioned groups? Did they support the research by providing the authors with a place to stay? Could you please clarify that?

Author Response

(The authors gave the same response as above.)
